# A Novel Secure End-to-End IoT Communication Scheme Using Lightweight Cryptography Based on Block Cipher

**Agus Winarno * and Riri Fitri Sari ***

Department of Electrical Engineering, University of Indonesia, Depok 16424, Indonesia
* Correspondence: agus.winarno01@ui.ac.id (A.W.); riri@ui.ac.id (R.F.S.)

**Abstract:** Personal data security is a cybersecurity trend that has captured the world's attention. Governments, practitioners and academics are jointly building personal data security in various communication systems, including IoT. The protocol that is widely used in IoT implementation is MQTT. By default, MQTT does not provide data security features in the form of data encryption. Therefore, this research was carried out on the design of Secure End-to-End Encryption MQTT with Block Cipher-Based Lightweight Cryptography. The protocol is designed by utilizing the Galantucci secret sharing scheme and a lightweight cryptographic algorithm based on a block cipher. The algorithms used include AES-128 GCM mode, GIFT-COFB, Romulus N1, and Tiny JAMBU. We tested the Secure End-to-End for MQTT protocol on the ARM M4 and ESP8266 processors. Our testing results on NodeMCU board, Tiny JAMBU have an average encryption time of 313 μs and an average decryption time of 327 μs. AES-128 GCM mode has an average encryption time of 572 μs and an average decryption time of 584 μs. GIFT-COFB has an average encryption time of 1094 μs and an average decryption time of 1110 μs. Meanwhile, Romulus N1 has an average encryption time of 2157 μs and an average decryption time of 2180 μs. On STM32L4 discovery, Tiny JAMBU had average encryption of 82 μs and an average decryption time of 85 μs. AES-128 GCM mode has an average encryption time of 163 μs and an average decryption time of 164 μs. GIFT-COFB has an average encryption time of 164 μs and an average decryption time of 165 μs. Meanwhile, Romulus N1 has an average encryption time of 605 μs and an average decryption time of 607 μs. Our experiment shows that the fastest performance is produced by Tiny JAMBU, followed by AES-128 Mode GCM, GIFT-COFB and Romulus N1.

**Keywords:** MQTT; block cipher; lightweight cryptography; secret sharing; IoT

## 1. Introduction

The implementation of 5G is a catalyst in the growth of the Internet of Things (IoT). Based on data from Ericson, the number of IoT devices connected to the internet network will reach 14.6 billion devices in 2021 and will increase to 30.2 billion devices in 2027 [1]. The McKinsey Global Institute also stated that the global IoT market could grow to 6.2 trillion dollars by 2025. The IoT market share is dominated by devices from the healthcare sector, which are worth 2.5 trillion dollars, and the manufacturing sector, which is worth around 2.3 trillion dollars [2,3]. By 2027, the 5G market potential for IoT devices will reach $297.8 billion [4]. The enormous IoT market also creates issues related to personal data protection [5–7]. Personal data protection is part of the goal of information security. Information security is a long and continuous process, not an instant result [8]. The fulfillment of information security services is a complex process that must be carried out carefully, starting from the establishment, implementation, and evaluation of information security policy compliance [9]. Effective information security must ensure a rational balance between requirements and controls provided to operate optimally and adequately [10]. Information security operates dynamically following technological developments, so it is necessary to find new methods through research channels, both offensive and defensive methods, in

accordance with applicable regulations [8]. This information security need must be met in the IoT communication protocol.

The communication protocol that is widely used in IoT implementations is MQTT [11]. By default, this protocol is not equipped with a data security system, so a security mechanism is needed for the protocol [12]. MQTT security can be achieved in the network [13], transport [14], and application [15]. Security at the user data layer can be achieved by encrypting the payload. In this research, a security design for the MQTT protocol is carried out using a Secure End-to-End Encryption MQTT with Block Cipher-Based Lightweight Cryptography. The use of end-to-end encryption is expected to be a data security solution at the application layer even though it uses an untrusted infrastructure.

Research related to MQTT security includes MQTT security at the application layer with the LBLOK, AES-128, PRESENT, Trivium, and Grain algorithms [16]. In addition, there is research on MQTT/MQTT-SN security with the ChaCha20-Poly1305 algorithm with AEAD mode [17] and MQTT security with a key delegation system for end-to-end encryption communication [18]. Each protocol is an MQTT security solution at the application layer.

In this study, Secure End-to-End Encryption on the MQTT protocol with Block Cipher-based Lightweight Cryptography was designed and implemented using the Galantucci secret sharing mechanism [19]. This secret sharing scheme is designed using multiple XOR operations, which are suitable for low-power IoT devices. The Secure End-to-End Encryption MQTT protocol design uses a lightweight cryptographic algorithm based on a block cipher, namely GIFT-COFB [20], Romulus [21,22], and Tiny JAMBU [23]. This scheme has several advantages:

1.  Secure End-to-End Encryption on the MQTT protocol with Block Cipher-based Lightweight Cryptography can be used for low power devices;
2.  This scheme provides convenience in key communication management by using the advantages of Galantucci secret sharing;
3.  It can be implemented for brokers who do not support the use of SSL/TLS or untrusted brokers.

The paper is structured as follows: Section 1 describes the introduction; Section 2 highlights related work; and Section 3 presents some of the literature reviews. Section 4 explains some experimental methods. Section 5 proposes the Secure End-to-End Encryption Scheme on the MQTT protocol with Block Cipher-based Lightweight Cryptography. Section 6 shows experimental results and discussion. Section 7 discusses our conclusions.

## 2. Related Works

The IoT has grown significantly in the industrial era 4.0. The utilization of the IoT has tremendous market potential, with an economic value of 5.5 trillion to 12.6 trillion dollars in 2030 [24]. In addition, IoT is also widely used for lifestyle fulfillment, process efficiency, and health. Privacy of data transmitted via IoT is of particular concern and handling [25]. Information security in transmission is an important aspect that must be considered.

MQTT is an IoT communication protocol that plays an important role in sending messages from publishers to subscribers. MQTT security can be done at the network, transport, and application levels. Research that has been done related to data security in the MQTT protocol at the application layer, among others, is the study of Peniak et al. [16] about secure communication models for IoT and MQTT. This research uses a symmetric key algorithm, namely PRESENT, LBLOK, AES-128, Trivium, and GRAIN. The use of symmetric keys, especially LWC in this study, is due to the consideration of limited memory usage on embedded devices [16].

A similar study was conducted by Sadio [17], regarding the end-to-end security scheme of the MQTT protocol using the ChaCha20-Po1y1305 algorithm with AEAD mode. The choice of this algorithm is due to the limitations of devices that do not support the use of TLS at the transport layer. ChaCha20-Po1y1305 is a popular lightweight stream cipher for devices with limited memory [17].

Research related to end-to-end encryption on MQTT was also carried out by Kumar et al. [18], who proposed an IoT security design in the form of **J**oining **E**ncryption and **D**elegation for **I**oT (JEDI). This scheme allows for end-to-end encryption with a key delegation system on each principal. Another feature that JEDI has is that it allows the encryption to be implemented in a decentralized key system, with decoupled communication and a tiered encryption system using the identity-based encryption algorithm with wildcard key derivation (WKD-IBE) [18].

Our research proposes a secure end-to-end encryption design for IoT communication protocols, especially MQTT, using a secret sharing scheme and a block cipher-based lightweight cryptography algorithm, i.e., AES-128 GCM mode, GIFT-COFB, ROMULUS N, and Tiny JAMBU. The secret sharing scheme used is secret sharing with the XOR model proposed by Galantucci et al. [19]. Table 1 describes the comparison of our research with previous research.

**Table 1.** Comparison with Previous Researchs.

| No | Research | Author | Algorithm |
|----|----------|--------|-----------|
| 1 | Extended Model of Secure Communication for Embedded Systems with IoT and MQTT | Peter Peniak et al. | LBLOK, AES-128, PRESENT, Trivium dan Grain |
| 2 | Lightweight Security Scheme for MQTT/MQTT-SN Protocol | Ousmane Sadio et al. | ChaCha20-Poly1305 |
| 3 | JEDI: Many-to-Many End-to-End Encryption and Key Delegation for IoT | Sam Kumar et al. | WKD-IBE |
| 4 | Secure End-to-End Encryption on the MQTT protocol with Block Cipher-based Lightweight Cryptography | Our Research | Secret Sharing, AES-128 GIFT-COFB, Romulus, dan Tiny JAMBU |

## 3. Background Theory

Section 3 will relay the background theory related to MQTT, Eclipse Mosquitto, Advanced Encryption Standard (AES), Lightweight Cryptography, BAN logic, and Secret Sharing.

### 3.1. Message Queuing Telemetry Transport (MQTT)

Message Queuing Telemetry Transport (MQTT) is an Internet of Things (IoT) protocol standard issued by OASIS [26]. MQTT is designed by carrying a publish/subscribe protocol that is lightweight and requires small bandwidth [27]. MQTT has been widely implemented in various industrial sectors, such as manufacturing, automotive, sports, energy, military, telecommunications, and health [28].

### 3.2. Eclipse Mosquitto Sebagai MQTT Broker

Eclipse Mosquitto is an open-source project that aims to provide client and server implementation features according to the MQTT standard. Mosquitto was developed by Roger A. Light with the aim of implementing IoT with the needs of light messaging, low-power devices, and limited network resources based on the MQTT protocol [29]. The development of Mosquitto is under the Eclipse Foundation.

Eclipse Mosquitto has been widely implemented in the academic world and in real life. Mosquitto can be used as a means of evaluating the MQTT protocol on railway bridge monitoring systems [30], data convergence in the electricity industry [31], and evaluating the performance of the publish-subscribe protocol on the IoT [32].

### 3.3. Advanced Encryption Standard (AES)

AES is a standard algorithm issued by NIST as a replacement for the Data Encryption Standard (DES). AES is designed as a symmetric key algorithm that manages messages in the form of blocks, often called block ciphers. NIST began announcing AES as a new standard starting from 26 November 2001 [33]. The algorithm chosen for AES is Rijndael, which was designed by Joan Daemen and Vincent Rijmen [34]. Rijndael was chosen as the

winner of a contest for the encryption standard algorithm held by NIST for 5 years. AES is designed to process data in blocks of 128 bits. AES has three key variations that can be used, namely AES-128, AES-192, and AES-256 [35].

Structurally, AES has a simple structure so that the encryption and decryption process can use the same structure. The AES process can be divided into three phases, namely the initial round, the main round, and the final round. In the initial round phase, the algorithm processes messages with the add round key function, namely the process of XORing messages with keys. In the next phase, AES process messages with the functions of sub bytes, shift rows, mix columns, and add round key. Meanwhile, the final round performs the same function as the main round but eliminates the mix column function [36]. The AES algorithm above has four main components, namely sub bytes, shift rows, mix columns, and add round key [37].

### 3.4. Lightweight Cryptography Final Candidate

Lightweight cryptography is an algorithm designed to meet the needs of cryptography in limited conditions that are not supported by current cryptography. In August 2018, NIST held a competition for the creation of lightweight cryptography standards. In the first round of the competition, there were 56 algorithm proposals submitted by researchers and practitioners. The algorithms were selected by NIST and became 32 algorithms in the second round and became 10 algorithms in the final round, which was announced in March 2021. The finalist candidates for lightweight cryptography included ASCON, Elephant, GIFT-COFB, Grain128-AEAD, ISAP, Photon- Beetle, Romulus, Sparkle, Tiny JAMBU, and Xoodyak. Of the 10 algorithms, there are three algorithms based on block ciphers, i.e., GIFT-COFB, Romulus, and Tiny JAMBU.

### 3.4.1. GIFT-COFB

The GIFT-COFB algorithm is a lightweight cryptography algorithm designed based on the GIFT block cipher algorithm with COMBined FeedBack (COFB) mode. The GIFT algorithm is an algorithm of the Substitution Permutation Network (SPN) type, which has two variants, namely GIFT 64-128 and GIFT-128-128. The two variants have differences in the length of the message block that is processed and the number of rounds. GIFT 64-128 can process 64-bit message blocks with a 128-bit key with 28 block iterations. Meanwhile, GIFT-128-128 can process 128-bit message blocks with 128-bit keys with 40 block iterations. In the GIFT-COFB algorithm, the variant used is GIFT-128-128 [20].

The COFB mode in the lightweight cryptography algorithm was designed by Chakraborti et al. with the aim of providing an authenticated encryption scheme. The COFB algorithm is an improvement algorithm of iCOFB to make it more efficient [38].

### 3.4.2. Tiny JAMBU

Tiny JAMBU was one of the finalists in the lightweight cryptography algorithm selection contest organized by NIST. Tiny JAMBU is designed based on the JAMBU block cipher by providing data authentication features. Tiny JAMBU has three key variants, namely 128-bit (Tiny JAMBU-128), 192-bit (Tiny JAMBU-192), and 256-bit (Tiny JAMBU-256). Tiny Jambu was designed by Hongjun Wu and Tao Huang by adding security parameters in the form of a 64-bit tag and a 96-bit nonce [23].

### 3.4.3. Romulus

Romulus is a lightweight cryptography mode AEAD algorithm based on Skinny 128/384+ Tweakable Block Ciphers. Romulus is designed to be optimally implemented on hardware with a security claim of 128-bit provable security [21,22]. Romulus has four variants, namely Romulus-N, Romulus-M, Romulus-T, and Romulus-H for hash functions.

### 3.5. BAN Logic

Burrows–Eternal–Needham (BAN) Logic is a formal testing method used to test protocols so that users can determine the security and trustworthiness of the information exchanged. This makes BAN logic often referred to as the logic of belief. BAN logic testing is carried out with the assumption that every exchange is carried out on an insecure medium and is susceptible to eavesdropping [39,40].

### 3.6. Secret Sharing

In this study, the use of secret sharing is aimed at supporting key management in the secure MQTT protocol. Secret sharing is a protocol scheme that supports multi-user communication by sharing sensitive information or a secret key into several parts. Such information can be recovered by a combination of part or all of the entity with a specified mechanism [41]. Secret sharing was first introduced by Adi Shamir with the aim of providing a solution for sharing secret keys in cryptographic communication [42].

Secret sharing protocols are widely used in our daily lives, such as in crypto currency wallets, for key management in vault by Hashicorp, and on Amazon Cloud HSM.

Secret sharing research has developed both in terms of method and efficiency. In 2012, Yi Sun et al. designed a secret sharing to simplify the reconstruction process without involving other entities [43]. In 2017, Kenko and Iwamura conducted a study to increase the efficiency of proactive secret sharing schemes in reducing communication traffic [44]. Another research project related to secret sharing was also carried out by Galantucci et al. regarding secret sharing with the XOR model, which is intended for distributed systems with many users [19]. This secret sharing scheme proposes a formal method, multiple XOR-ED secret sharing approach, and One Time User Key.

## 4. Method

The research flow used in the research includes problem identification, literature study, design and implementation, testing and analysis, and conclusion. Our research flow can be seen in Figure 1.

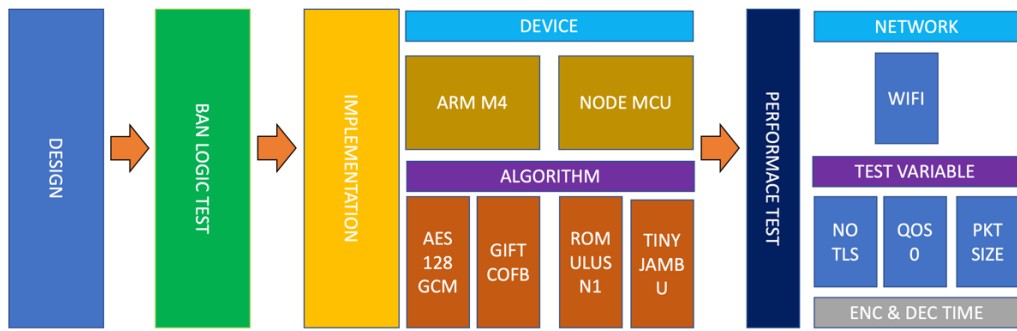

**Figure 1.** Research Flow.

We designed a secure end-to-end encryption for IoT communications by leveraging Galatucci's secret sharing as a key management scheme for generation and rekeying. The algorithm used in the design is a final candidate for lightweight cryptography based on block ciphers. Implementation is carried out at the communication stage on ARM M4 and NodeMCU.

The lightweight algorithm performance test was carried out on WiFi networks. Performance testing parameters include encryption, decryption, and publication of messages to the MQTT broker.

## 5. Secure End-to-End Encryption for Iot Communication Scheme

The design of the secure MQTT protocol aims to provide information security guarantees in the form of data confidentiality and authentication. This design can be applied

to hardware devices that do not support SSL/TLS so that they can communicate securely even through the internet. The general design of the protocol can be seen in Figure 2.

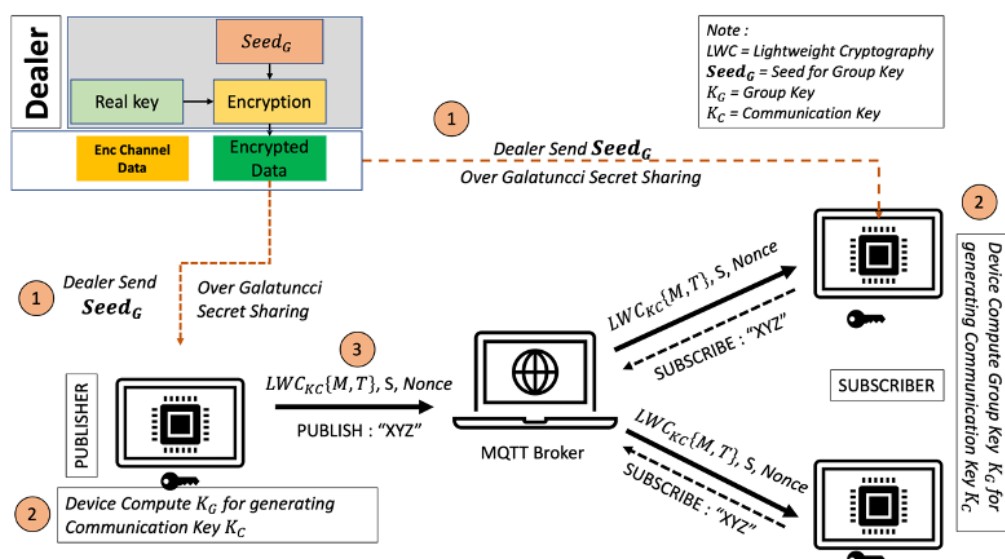

**Figure 2.** Secure End-to-End Design on MQTT Protocol.

Figure 2 provides an explanation of the working system of the Secure End-to-End Scheme on the MQTT Protocol. Here is the explanation:

1.  The dealer sends a *seed*$_G$ via Galatuncci's secret sharing as a component in calculating the group key $K_G$;
2.  Each device that receives the seed$_G$ can compute the group key $K_G$. Each device that is registered and has the same license will have the same group key $K_G$;
3.  Each device that has the same $K_G$ group key can communicate with the session key $K_C$ derived from the group key $K_G$. The cryptographic algorithm used in communication is lightweight cryptography (LWC), which provides data confidentiality and authentication features.

Secure end-to-end MQTT with lightweight cryptography designed based on Galatuncci's secret sharing scheme [19] and lightweight cryptography. The secure end-to-end MQTT protocol was designed with the aim of providing encryption for message payloads sent over the MQTT protocol. The secret sharing scheme used is intended to make it easier for dealers to generate group keys, device management, and key updates. Updates are made if there is an addition or deletion of devices.

The secure end-to-end MQTT design is divided into three stages, i.e., registration and key generation, communication, and rekeying.

### 5.1. Registration and Key Generation

The registration process for the secure end-to-end MQTT protocol is the initial phase to register devices that communicate in the MQTT protocol. This phase is controlled by a factory key ($K_F$) and a subscription key ($K_S$). The factory key in the protocol is used for communication between dealers or third parties with IoT devices in a secret sharing scheme to generate group key seeds. Meanwhile, the subscription key functions as the master key in generating the group key in combination with the group key seed seed$_G$.

The registration phase of the secure end-to-end MQTT protocol is designed based on the secret sharing protocol developed by Galantucci et al. [19]. This protocol was chosen because it supports multi-user communication using light operations because it uses XOR operations. This scheme also has advantages in several key management schemes. Implementation of Galantucci's secret sharing scheme can be intended for several uses, including one time keys, groups of valid keys, and dealer key management.

The pre-registration phase aims to encrypt the channel before the registration as shown in the following process:

1. Set-up environment:
   a. Dealers store information on salt values, factory key hashes, and device subscription keys.
   b. Each device has a factory key and a subscription key and can access salt information.
   c. The dealer and each device have the functions, $\Omega$ and $\omega$, which satisfy Galantucci's secret sharing rule [19].

2. Encrypted Channel Creation Process
   a. The dealer creates a one time user key for the IoT device and himself. One time user keys (OTUK) of IoT devices can be generated from the hash value of the factory key and device salt stored at the dealer.
   b. Dealers generate communication keys and encrypted channels from dealer OTUK and OTUK IoT devices.

3. Process on IoT Device
   a. The IoT device creates a device OTUK from the factory key and salt provided by the dealer.
   b. IoT devices can communicate securely with encrypted channels created by dealers.
   c. Through this encrypted channel, the device can exchange information or key seeds securely with the dealer.

The proposed registration scheme for Galantucci's secret sharing is as a secure end-to-end MQTT registration scheme as shown in Figure 3.

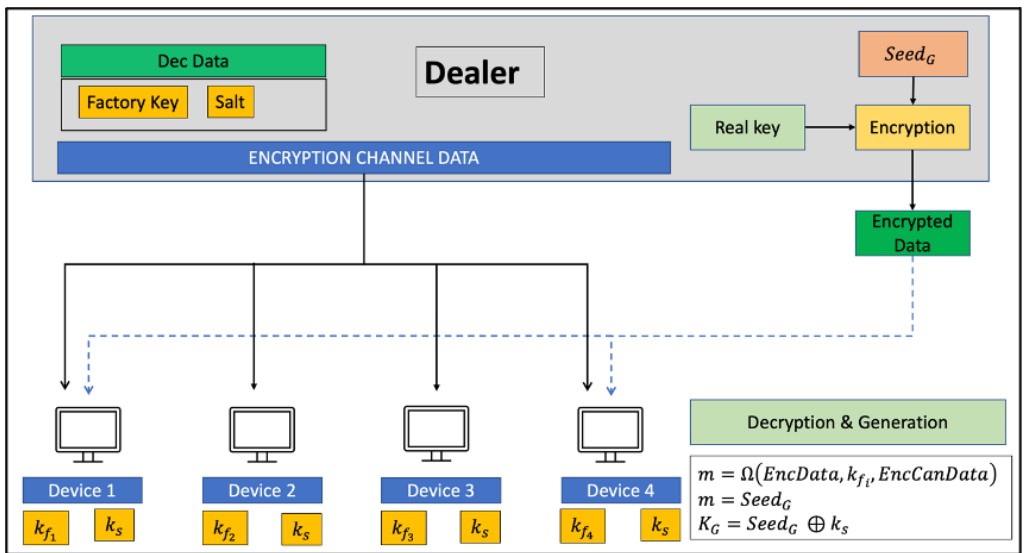

**Figure 3.** Group Key Generation Phase.

At this registration stage, each IoT device group key ($K_G$) is the same to communicate in the MQTT protocol.

### 5.2. Communication

This phase is an important part of the protocol in the communication process between devices in the MQTT network. In this phase, all IoT devices that have the same group key ($K_G$) can communicate and exchange information securely. The following is the proposed communication protocol:

1. Set-up environment:
   a. Device A and Device B have Key Group $K_G$
   b. Device A generates a Nonce to generate a session key $K_C$ as communication key
2. Encrypted Channel Creation Process
   a. Calculates the session key ($K_C$) used for communication. $K_C = K_G \oplus Nonce$
   b. Calculating the S signature to check the validity of the communication key $K_C$:
      $S = hash\ (Nonce||hash(KG)^{128})$
   c. Device A sends an encrypted message C to B via MQTT Broker: $E_{KC}\{M,\ t_A\}, S,$ *Nonce*
3. Device B receives encrypted messages from Device A via the MQTT broker. Device B does calculate the following:
   a. Calculating the Signature $S'$ with the nonce information sent and $K_G$ held:
      $S' = hash\ (Nonce||hash(KG)^{128})$
   b. If $S' \neq S$, the message is not processed. However, if $S' \neq S$, then go to step c.
   c. Performs session key calculations for communication with received nonce information. $K_C = K_G \oplus Nonce$
   d. Decrypt the message with $K_C$ calculated to get the message. $D_{KC}\{E_{KC}\{M,\ t_A\}\}$
   e. Device B gets messages M and $t_A$, the information $t_A$ is used to check the freshness of the received message.

The protocol in the communication phase above can be visualized with a communication scheme as shown in Figure 4.

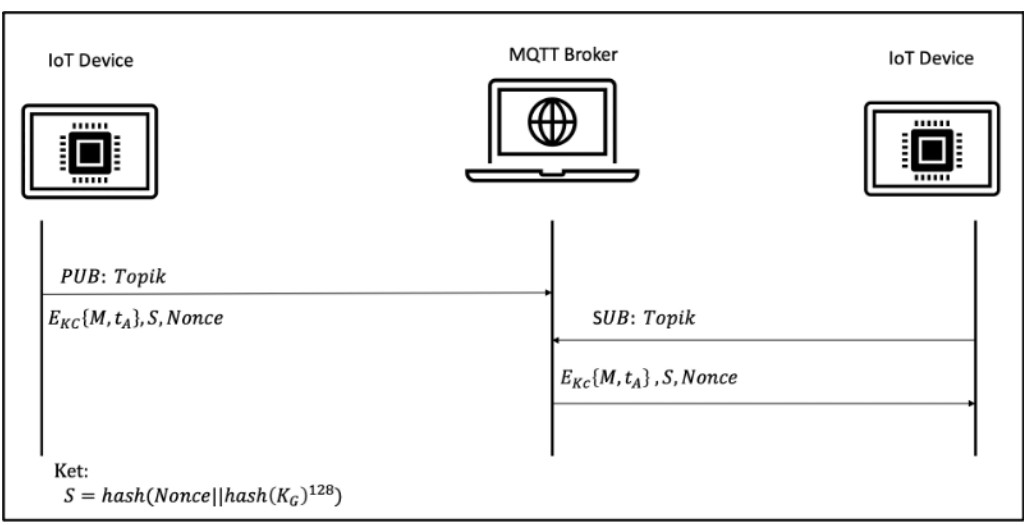

**Figure 4.** Communication Phase.

*5.3. Rekeying*

The key update process is a control in key management on the protocol in the use and renewal of group keys (group key). This phase makes it easy for the system to update the key when there is an addition or deletion of devices connected to the communication scheme. The key update scheme in this protocol takes advantage of the easy key management of the secret sharing scheme designed by Galantucci et al. [19]. The key update scheme for secure end-to-end on the MQTT protocol is as follows:

- The system is experiencing an increase or decrease in the number of IoT devices.
- Dealers update legitimate devices connected to the network
- The dealer updates the encrypted channel data and seed to generate the group key ($K_G$) with the mechanism in Section 5.1.
- The dealer sends the seed to the legitimate device.

- Each legitimate device receives the seed and recalculates $K_G$ as the group key to communicate.

## 6. Results

This section may be divided by subheadings. It should provide a concise and precise description of the experimental results, their interpretation, as well as the experimental conclusions that can be drawn.

### 6.1. BAN Logic Test

Our testing for the design of the secure MQTT protocol was carried out using the BAN logic method. The test is carried out based on the rules that apply in BAN logic, namely the message meaning rule, the non-verification rule, and the jurisdiction rule. The stages carried out in the BAN logic test consist of four steps, namely:

- Changing the protocol in its ideal form;
- Making assumptions from the ideal protocol;
- Explain each step in the ideal protocol;
- Carry out protocol evaluation with BAN logic rules.

The communication protocol for secure end-to-end encryption on the MQTT protocol has the following form:

$$A \rightarrow B : E_{KC}\{M, Nonce_A\}, S, Nonce \tag{1}$$

The steps that must be taken in testing the BAN logic in the secure MQTT design are:

1.  Changing the protocol in the ideal form:

$$M1 \; A \rightarrow B : \left\{ \left( A \overset{K_{ab}}{\leftrightarrow} B \right), N_A \right\}_{K_{ab}}, \sharp \left\{ A \overset{K_{ab}}{\leftrightarrow} B \right\}_{K_{ab}} \tag{2}$$

2.  Making assumptions from the ideal protocol:

$$P1 \; A \models A \overset{K_{ab}}{\leftrightarrow} B \tag{3}$$

$$P2 \; B \models A \overset{K_{ab}}{\leftrightarrow} B \tag{4}$$

$$P3 \; B \models A \Rightarrow A \overset{K_{ab}}{\leftrightarrow} B \tag{5}$$

$$P4 \; B \models \left( A \Rightarrow \sharp (A \overset{K_{ab}}{\leftrightarrow} B) \right) \tag{6}$$

$$P5 \; A \models \sharp (N_A) \tag{7}$$

3.  Explain each step in the ideal protocol:

$$P6 \; B \lhd \{N_A\}_{K_{ab}} \tag{8}$$

4.  Carry out protocol evaluation with BAN logic rules.

    After M1

    This test aims to prove that party $B$ believes in the key to the communication between $A$ and $B$, as well as the novelty of the key used.

- Jurisdiction rule

$$\frac{B \models A \Rightarrow (A \overset{K_{ab}}{\leftrightarrow} B), B \models A \models (A \overset{K_{ab}}{\leftrightarrow} B)}{B \models (A \overset{K_{ab}}{\leftrightarrow} B)} \tag{9}$$

$$\frac{B \mathrel{|\!\equiv} \left( A \Rightarrow \sharp(A \overset{K_{ab}}{\leftrightarrow} B) \right), B \mathrel{|\!\equiv} A \left( A \mathrel{|\!\equiv} \sharp(A \overset{K_{ab}}{\leftrightarrow} B) \right)}{B \mathrel{|\!\equiv} \sharp(A \overset{K_{ab}}{\leftrightarrow} B)} \tag{10}$$

After *B* trusts the communication key between *A* and *B*, BAN logic proves that the message sent is a new message through the trust nonce sent by *A*.

- Message meaning rule

$$\frac{B \mathrel{|\!\equiv} A \Rightarrow A \overset{K_{ab}}{\leftrightarrow} B, B \mathrel{\triangleleft} \{N_A\}_{K_{ab}}}{B \mathrel{|\!\equiv} A \mathrel{|\!\sim} N_A} \tag{11}$$

- Nonce Verification rule

$$\frac{A \Rightarrow \sharp(N_A), B \mathrel{|\!\equiv} A \mathrel{|\!\sim} N_A}{B \mathrel{|\!\equiv} A \mathrel{|\!\equiv} N_A} \tag{12}$$

- Jurisdiction rule

$$\frac{B \mathrel{|\!\equiv} A \Rightarrow N_A, B \mathrel{|\!\equiv} A \mathrel{|\!\equiv} N_A}{B \mathrel{|\!\equiv} N_A} \tag{13}$$

In the BAN logic test, the two entities can trust each other. Party *B* believes that party *A* has the same access rights and group key. Party *B* believes that the message sent by *B* is a fresh message.

### 6.2. Performance on ESP 8266 and ARM M4

The process of implementing secure end-to-end encryption on the MQTT protocol is carried out on two hardware devices, namely STM32L4 Discovery (ARM M4) and NodeMCU (esp 8266). Both hardware are low-power microcontrollers that are suitable for IoT applications. In the implementation of the protocol, it is only carried out at the communication stage.

At the design stage, the package sent from publisher to subscriber is $E_{KC}\{M, Nonce_A\}$, $S, Nonce$. At the implementation stage, $S$ and $Nonce$ are ignored to check the performance of the implemented cryptographic algorithms. The implemented communication design can be seen in Figure 5.

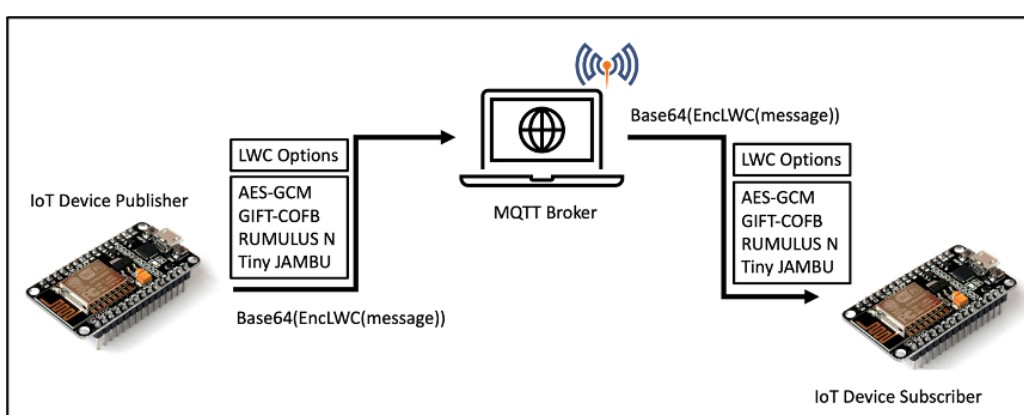

**Figure 5.** Secure MQTT Implementation Scheme with LWC.

In the process of implementing secure end-to-end encryption, the MQTT protocol utilizes a lightweight cryptographic algorithm that has features that can guarantee the confidentiality and authenticity of data. Lightweight cryptographic algorithms implemented in this study include AES-128 GCM mode, GIFT COFB, ROMULUS N, and Tiny JAMBU.

These four algorithms are the final candidates in the lightweight cryptographic algorithm competition organized by NIST.

Testing the performance of the lightweight cryptography algorithm on STM32L4 and ESP8266 is done by setting several test parameters, namely encryption time, decryption time, and the time required for packet delivery. The test is carried out on a WiFi network with specifications as shown in Table 2:

**Table 2.** Test Parameters.

| No | Parameter | Description | Value |
|----|-----------|-------------|-------|
| 1 | WiFi | Download Speed | 43.49 Mbps |
| | | Upload Speed | 46.73 Mbps |
| 2 | Messages | 80 bit (10 Char) | hello eqmx |
| | | 120 bit (10 Char) | hello eqmx dear |
| | | 160 bit (10 Char) | hello eqmx dear agus |

Based on Figure 5, encrypted messages are sent with base64 and encoded so that the encrypted messages are not changed during transmission from publisher to subscriber. Sending encrypted messages without base64 has the potential that the message cannot be decrypted because the encrypted message can be in the form of special Characters that can affect the transmission process, such as Null, Delete, or Enter Characters. The MQTT server used in this simulation is Eclipse Mosquitto version 2. This MQTT server supports MQTT version 5. In this study, the QOS used by the publisher is QOS0. The results of testing the secure MQTT protocol are presented in Sections 6.2.1 and 6.2.2.

### 6.2.1. Node MCU

The performance test of the lightweight encryption algorithm using anonymous and registered accounts on the MQTT server was carried out on the WiFi network. The registered account used for testing is "winarnolab" with the password "loop1206". The test was carried out with three different message lengths, namely 10, 15, and 20 Characters.

- Anonymous Account

The results of the tests carried out on the WiFi network as anonymous users can be seen in Tables 3–5 and Figure 6. Testing is done by sending messages with a length of 80 bits (10 Characters), 120 bits (15 Characters), and 160 bits (20 Characters).

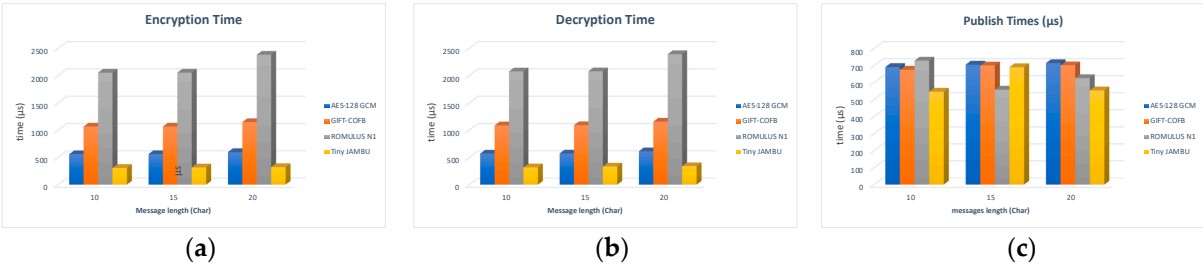

**Figure 6.** Lightweight Cryptographic Performance on NodeMCU on WiFi Network Used Anonymous Account. (**a**) Encryption Time; (**b**) Decryption Time; (**c**) Message Publication Time to MQTT Server.

**Table 3.** Lightweight Cryptographic Encryption and Decryption Performance on NodeMCU with 10 Characters Messages used Anonymous Account.

| No | LWC Algorithm | Encryption Time (μs) | Decryption Time (μs) | Publication Time (μs) |
|---|---|---|---|---|
| 1 | AES-128 GCM | 558 | 570 | 690 |
| 2 | GIFT-COFB | 1066 | 1087 | 674 |
| 3 | Tiny JAMBU | 306 | 317 | 545 |
| 4 | Romulus-N1 | 2052 | 2075 | 728 |

**Table 4.** Lightweight Cryptographic Encryption and Decryption Performance on NodeMCU with 15 Characters Messages used Anonymous Account.

| No | LWC Algorithm | Encryption Time (μs) | Decryption Time (μs) | Publication Time (μs) |
|---|---|---|---|---|
| 1 | AES-128 GCM | 560 | 571 | 704 |
| 2 | GIFT-COFB | 1066 | 1090 | 699 |
| 3 | Tiny JAMBU | 315 | 332 | 557 |
| 4 | Romulus-N1 | 2053 | 2077 | 689 |

**Table 5.** Lightweight Cryptographic Encryption and Decryption Performance on NodeMCU with 20 Characters Messages used Anonymous Account.

| No | LWC Algorithm | Encryption Time (μs) | Decryption Time (μs) | Publication Time (μs) |
|---|---|---|---|---|
| 1 | AES-128 GCM | 596 | 612 | 713 |
| 2 | GIFT-COFB | 1146 | 1156 | 700 |
| 3 | Tiny JAMBU | 321 | 338 | 553 |
| 4 | Romulus-N1 | 2381 | 2391 | 625 |

From Table 3, it can be seen that Tiny JAMBU can encrypt messages with a time of 306 μs and decryption of 317 μs, for messages with a length of 10 Characters. AES-128 GCM mode can perform encryption and decryption of 558 μs and 570 μs, respectively. GIFT-COFB takes 1066 μs and 1087 μs for encryption and decryption. Meanwhile, Romulus N1 is capable of encrypting 10-character messages for 2052 μs and 2075 μs and message publication to the MQTT server takes 728 μs.

Table 4 shows the results of testing the secure end-to-end encryption on the MQTT protocol by sending a 15-character message on a WiFi network as an anonymous user on the MQTT server. Tiny JAMBU produces an excellent lightweight cryptographic algorithm with an encryption time of 315 μs, a decryption time of 332 μs, and a message publication time of 557 μs. Meanwhile, the AES-128 GCM is still tight with an encryption time of 560 μs and a decryption time of 571 μs and a publication time of 704 μs. This is followed by GIFT-COFB with an encryption time of 1066 μs and a decryption time of 1090 μs and Romulus N1 with an encryption time of 2053 μs and a decryption time of 2077 μs with a message publication time of 689 μs.

Table 5 shows the results of testing secure end-to-end encryption on the MQTT protocol by sending messages of 20 Characters long on the WiFi network as an anonymous user on the MQTT server. In the results of this test, the highest performance is still produced by Tiny JAMBU, with an encryption time of 321 μs and a decryption time of 338 μs.

Based on the results of testing secure end-to-end encryption on the MQTT protocol on the WiFi network as an anonymous user of the MQTT server, it was discovered that sending messages of 10 to 20 Characters can be done in less than 1 ms and the fastest performance is produced by Tiny JAMBU, as can be seen in Figure 6.

- Registered account

Testing of secure end-to-end encryption on the MQTT protocol on a WiFi network as a registered user of the MQTT server can be seen in Tables 6–8 and Figure 7. Testing is done by sending messages with a length of 80 bits (10 Characters), 120 bits (15 Characters) and 160 bits (20 Characters).

**Table 6.** Lightweight Cryptographic Encryption and Decryption Performance on NodeMCU with 10 Characters Messages used Registered Account.

| No | LWC Algorithm | Encryption Time (μs) | Decryption Time (μs) | Publication Time (μs) |
|----|---------------|----------------------|----------------------|------------------------|
| 1 | AES-128 GCM | 559 | 569 | 679 |
| 2 | GIFT-COFB | 1066 | 1084 | 674 |
| 3 | Tiny JAMBU | 305 | 317 | 541 |
| 4 | Romulus-N1 | 2051 | 2071 | 664 |

**Table 7.** Lightweight Cryptographic Encryption and Decryption Performance on NodeMCU with 15 Characters Messages used Registered Account.

| No | LWC Algorithm | Encryption Time (μs) | Decryption Time (μs) | Publication Time (μs) |
|----|---------------|----------------------|----------------------|------------------------|
| 1 | AES-128 GCM | 559 | 572 | 697 |
| 2 | GIFT-COFB | 1066 | 1090 | 688 |
| 3 | Tiny JAMBU | 315 | 332 | 543 |
| 4 | Romulus-N1 | 2052 | 2078 | 680 |

**Table 8.** Lightweight Cryptographic Encryption and Decryption Performance on NodeMCU with 20 Characters Messages used Registered Account.

| No | LWC Algorithm | Encryption Time (μs) | Decryption Time (μs) | Publication Time (μs) |
|----|---------------|----------------------|----------------------|------------------------|
| 1 | AES-128 GCM | 596 | 608 | 693 |
| 2 | GIFT-COFB | 1146 | 1160 | 705 |
| 3 | Tiny JAMBU | 321 | 334 | 545 |
| 4 | Romulus-N1 | 2365 | 2391 | 689 |

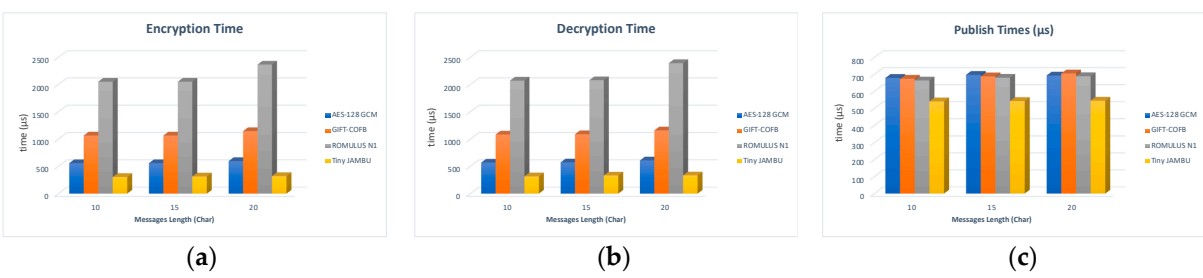

**Figure 7.** Lightweight Cryptographic Performance on NodeMCU on WiFi Network Used Registered Account. (**a**) Encryption Time; (**b**) Decryption Time; (**c**) Message Publication Time to MQTT Server.

From Table 6, it can be seen that Tiny JAMBU is able to encrypt messages with a time of 305 μs and decryption of 317 μs for messages with a length of 10 Characters. GIFT-COFB takes 1066 μs and 1084 μs for encryption and decryption. AES-128 GCM mode can perform encryption and decryption with a time of 559 μs and 569 μs. Meanwhile, Romulus N1 can encrypt 10-character messages for 2051 μs and 2071 μs.

Table 7 provides an overview of the performance of lightweight cryptographic algorithms on secure MQTT on a WiFi network. The test results show tiny JAMBU has the

fastest performance, with encryption and decryption times of 315 μs and 332 μs, respectively. Meanwhile, the lowest performance was produced by Romulus-N1, 2053 *μs* for encryption and 2078 μs for decryption.

Table 8 shows that tiny JAMBU and AES-128 GCM Mode can perform encryption and decryption in less than 1 ms. Tiny JAMBU can encrypt for 315 μs and decrypt for 332 μs. AES-128 GCM mode requires 596 μs for encryption and 612 μs for decryption. Meanwhile, GIFT-COFB and Romulus-N1 can perform encryption and decryption in 1 ms and 2 ms, respectively.

Figure 7 provides information that sending messages of 10 to 20 Characters in the secure MQTT test on a WiFi network as a registered user of the MQTT server can be carried out in less than 1 ms and the fastest performance is produced by Tiny JAMBU with a time of 545 μs.

### 6.2.2. STM32L4 Discovery

The performance test of the lightweight encryption algorithm using an anonymous and a registered account on the MQTT server was carried out on WiFi network. The registered account used for testing is "winarnolab" with the password "loop1206". The test is carried out with three different message lengths, namely 10, 15, and 20 Characters.

- Anonymous Account

The results of the tests carried out on the WiFi network as anonymous users can be seen in Tables 9–11, and Figure 8.

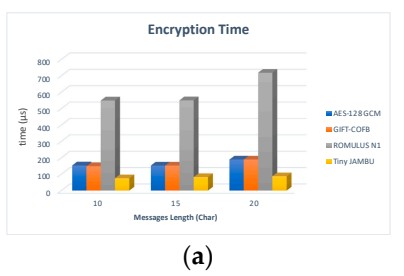
(**a**)

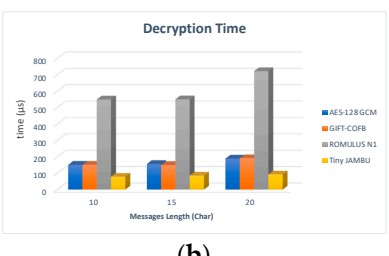
(**b**)

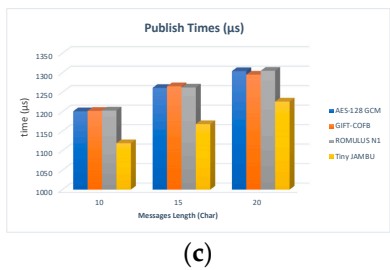
(**c**)

**Figure 8.** Lightweight Cryptographic Encryption and Decryption Performance on STM32L4 on WiFi Network Used Anonymous Account. (**a**) Encryption Time; (**b**) Decryption Time; (**c**) Message Publication Time to MQTT Server.

**Table 9.** Lightweight Cryptographic Encryption and Decryption Performance on STM32L4 with 10 Characters Messages Used Anonymous Account.

| No | LWC Algorithm | Encryption Time (μs) | Decryption Time (μs) | Publication Time (μs) |
|----|---------------|----------------------|----------------------|-----------------------|
| 1 | AES-128 GCM | 153 | 151 | 1201 |
| 2 | GIFT-COFB | 149 | 151 | 1202 |
| 3 | Tiny JAMBU | 75 | 78 | 1119 |
| 4 | Romulus-N1 | 549 | 549 | 1203 |

**Table 10.** Lightweight Cryptographic Encryption and Decryption Performance on STM32L4 with 15 Characters Messages Used Anonymous Account.

| No | LWC Algorithm | Encryption Time (μs) | Decryption Time (μs) | Publication Time (μs) |
|----|---------------|----------------------|----------------------|-----------------------|
| 1 | AES-128 GCM | 152 | 156 | 1261 |
| 2 | GIFT-COFB | 152 | 150 | 1265 |
| 3 | Tiny JAMBU | 82 | 85 | 1168 |
| 4 | Romulus-N1 | 550 | 550 | 1262 |

**Table 11.** Lightweight Cryptographic Encryption and Decryption Performance on STM32L4 with 20 Characters Messages Used Anonymous Account.

| No | LWC Algorithm | Encryption Time (µs) | Decryption Time (µs) | Publication Time (µs) |
|---|---|---|---|---|
| 1 | AES-128 GCM | 189 | 189 | 1304 |
| 2 | GIFT-COFB | 189 | 191 | 1295 |
| 3 | Tiny JAMBU | 87 | 92 | 1226 |
| 4 | Romulus-N1 | 718 | 722 | 1305 |

From Table 9, it can be seen that Tiny JAMBU can encrypt messages with a time of 75 µs and decryption of 78 µs, for messages with a length of 10 Characters. AES-128 GCM can perform encryption and decryption with 153 µs, and 151 µs. GIFT-COFB takes 149 µs and 151 µs for encryption and decryption. Meanwhile, Romulus N1 can encrypt 10-character messages for 549 µs and 549 µs and message publication to the MQTT server takes 1203 µs.

Table 10 shows the results of testing the secure end-to-end encryption on the MQTT protocol by sending a 15-character message on a WiFi network as an anonymous user on the MQTT server. Tiny JAMBU produces an excellent lightweight cryptographic algorithm with an encryption time of 82 µs, a decryption time of 85 µs, and a message publication time of 1168 µs.

Meanwhile, the AES-128 GCM is still tight with an encryption time of 152 µs and a decryption time of 156 µs and a publication time of 1261 µs. This is followed by GIFT-COFB with an encryption time of 152 µs and a decryption time of 150 µs and Romulus N1 with an encryption time of 550 µs and a decryption time of 550 µs with a message publication time of 1262 µs.

Table 11 shows the results of testing secure end-to-end encryption on the MQTT protocol by sending messages of 20 Characters long on the WiFi network as an anonymous user on the MQTT server. In the results of this test, the highest performance is produced by Tiny JAMBU with an encryption time of 87 µs and a decryption time of 92 µs.

Based on the results of testing secure end-to-end encryption on the MQTT protocol on the WiFi network as an anonymous user of the MQTT server, it was discovered that sending messages of 10 to 20 Characters can be done in less than 1.4 ms and the fastest performance is produced by Tiny JAMBU, as can be seen in Figure 8.

- Registered Account

Testing of secure end-to-end encryption on the MQTT protocol on a WiFi network as a registered user of the MQTT server can be seen in Tables 12–14 and Figure 9. Testing is done by sending messages with a length of 80 bits (10 Characters), 120 bits (15 Characters), and 160 bits (20 Characters).

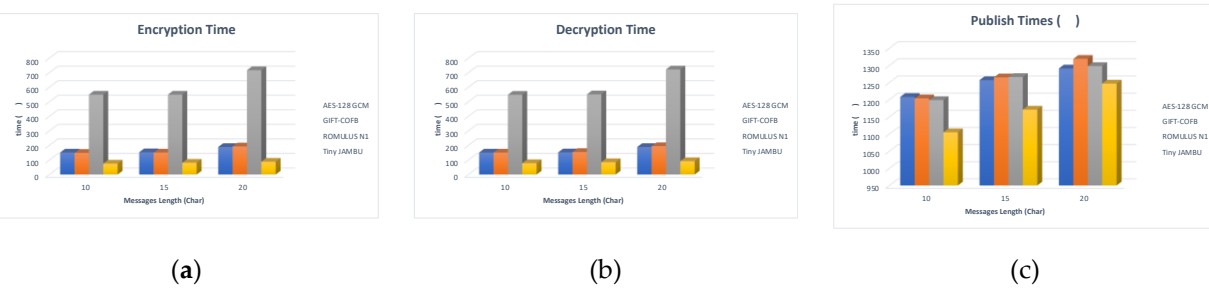

(**a**)             (**b**)             (**c**)

**Figure 9.** Lightweight Cryptographic Encryption and Decryption Performance on STM32L4 on WiFi Network Used Registered Account. (**a**) Encryption Time; (**b**) Decryption Time; (**c**) Message Publication Time to MQTT Server.

**Table 12.** Lightweight Cryptographic Encryption and Decryption Performance on STM32L4 with 10 Characters Messages Used Registered Account.

| No | LWC Algorithm | Encryption Time (μs) | Decryption Time (μs) | Publication Time (μs) |
|----|---------------|----------------------|----------------------|-----------------------|
| 1 | AES-128 GCM | 151 | 151 | 1208 |
| 2 | GIFT-COFB | 149 | 151 | 1204 |
| 3 | Tiny JAMBU | 75 | 78 | 1105 |
| 4 | Romulus-N1 | 549 | 548 | 1199 |

**Table 13.** Lightweight Cryptographic Encryption and Decryption Performance on STM32L4 with 15 Characters Messages Used Registered Account.

| No | LWC Algorithm | Encryption Time (μs) | Decryption Time (μs) | Publication Time (μs) |
|----|---------------|----------------------|----------------------|-----------------------|
| 1 | AES-128 GCM | 153 | 152 | 1257 |
| 2 | GIFT-COFB | 151 | 155 | 1265 |
| 3 | Tiny JAMBU | 81 | 85 | 1171 |
| 4 | Romulus-N1 | 549 | 551 | 1266 |

**Table 14.** Lightweight Cryptographic Encryption and Decryption Performance on STM32L4 with 20 Characters Messages Used Registered Account.

| No | LWC Algorithm | Encryption Time (μs) | Decryption Time (μs) | Publication Time (μs) |
|----|---------------|----------------------|----------------------|-----------------------|
| 1 | AES-128 GCM | 189 | 189 | 1291 |
| 2 | GIFT-COFB | 193 | 195 | 1319 |
| 3 | Tiny JAMBU | 88 | 92 | 1247 |
| 4 | Romulus-N1 | 717 | 722 | 1298 |

In Table 12, it can be seen that Tiny JAMBU is able to encrypt messages with a time of 75 μs and decryption of 78 μs for messages with a length of 10 Characters. GIFT-COFB takes 149 μs and 151 μs for encryption and decryption. AES-128 GCM mode can perform encryption and decryption with a time of 151 μs and 151 μs. Meanwhile, Romulus N1 can encrypt 10-character messages for 549 μs. and 548 μs.

Table 13 provides an overview of the performance of lightweight cryptographic algorithms on secure MQTT on a WiFi network. The test results show tiny JAMBU has the fastest performance with encryption and decryption time of 81 μs and 85 μs, respectively. Meanwhile, the lowest performance was produced by Romulus-N1, 549 μs for encryption and 551 μs for decryption.

Table 14 shows that tiny JAMBU can perform encryption and decryption in less than 0.1 ms. Tiny JAMBU can encrypt for 88 μs and decrypt for 92 μs. AES-128 GCM and GIFT-COFB can perform encryption and decryption in less than 0.2 ms. AES-128 GCM requires 189 μs for encryption and 189 μs for decryption. GIFT-COFB requires 193 μs for encryption and 195 μs for decryption Meanwhile, Romulus-N1 can perform encryption and decryption in 717 μs and 722 μs.

Based on the results of testing secure end-to-end encryption on the MQTT protocol on a WiFi network as a registered user of the MQTT server, information is obtained that sending messages of 10 to 20 Characters can be done from 1.1 ms to 1.3 ms and the fastest performance is produced by tiny JAMBU with a publication time of 1105 μs to 1247 μs, as can be seen in Figure 9.

## 7. Conclusions

MQTT Secure End-to-End Encryption Design with Lightweight Cryptography Based on Block Cipher is designed by utilizing Galantucci's secret sharing, which makes it easy to perform key and user management and uses multiple XOR operations that are compatible with low-power IoT devices.

The performance of Secure End-to-End Encryption MQTT with Block Cipher-Based Lightweight Cryptography is performed on two development boards, i.e., NodeMCU and STM32L4 Discovery. The result of our testing in NodeMCU, the Tiny JAMBU algorithm can perform encryption for 313 µs and decryption for 327 µs. AES-128 GCM mode can encrypt for 572 µs and decrypt for 584 µs, while GIFT-COFB can encrypt for 1094 µs and decrypt for 1110 µs. Meanwhile, Romulus N1 has an encryption time of 2157 µs and a decryption time of 2180 µs. In STM32L4 Discovery, the Tiny JAMBU algorithm can perform encryption for 82 µs and decryption for 85 µs. AES-128 GCM mode encrypts for 163 µs and decrypts for 164 µs. GIFT-COFB can encrypt for 164 µs and decrypt for 165 µs. Romulus N1 encrypts for 605 µs and decrypts for 607 µs.

**Author Contributions:** A.W. and R.F.S., overall design scheme; A.W., implemented the communication phase in STM32L4 and NodeMCU; A.W. and R.F.S., conducted tests and data analysis; A.W. and R.F.S., wrote the paper. All authors have read and agreed to the published version of the manuscript.

**Funding:** This work is supported by Universitas Indonesia under PUTI Q2 Grant with contract number NKB-710/UN2.RST/HKP.05.00/2022.

**Institutional Review Board Statement:** Not applicable.

**Informed Consent Statement:** Not applicable.

**Data Availability Statement:** The data presented in this study are available from the corresponding author upon request.

**Acknowledgments:** This work is supported by Universitas Indonesia.

**Conflicts of Interest:** The authors declare no conflict of interest.

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
