# Peer review of "A Novel Secure End-to-End IoT Communication Scheme Using Lightweight Cryptography Based on Block Cipher"

_applsci, doi:10.3390/app12178817_

Round 1

Reviewer 1 Report

The paper proposes an interesting method for secure end-to-end IoT communications. The paper is well organized and written. I have a comment.

-       All comparisons are done in sense of time and complexity. I suggest that the authors also compare different schemes in sense of security.

Author Response

The following is a list of revisions that have been made:
1. Added information security explanation on Line No. 40 to 49
2. Adding the advantages of the Secure End-to-End Encryption scheme on the MQTT protocol with Block Cipher-based Light-weight Cryptography Line No 69 to 76
3. Replace the word explanation in The paper is structured Line No. 77 to 81
4. Adding a table containing comparisons with previous research Line No 114
5. Adding a description to the conclusion of lines no. 599 to 608

Reviewer 2 Report

The manuscript is well organized and writed. 

Author Response

(The authors gave the same response as above.)

Author Response

(The authors gave the same response as above.)
